# Reduced-Beclin1-Expressing Mice Infected with Zika-R103451 and Viral-Associated Pathology during Pregnancy

**DOI:** 10.3390/v12060608

**Published:** 2020-06-02

**Authors:** Mohan Kumar Muthu Karuppan, Chet Raj Ojha, Myosotys Rodriguez, Jessica Lapierre, M. Javad Aman, Fatah Kashanchi, Michal Toborek, Madhavan Nair, Nazira El-Hage

**Affiliations:** 1Department of Immunology and Nanomedicine, Herbert Wertheim College of Medicine, Florida International University, Miami, FL 33199, USA; mmuthuka@fiu.edu (M.K.M.K.); cojha001@fiu.edu (C.R.O.); myrodrig@fiu.edu (M.R.); jlapi008@fiu.edu (J.L.); nairm@fiu.edu (M.N.); 2Integrated Biotherapeutics, Rockville, MD 20850, USA; Jaman@integratedbiotherapeutics.com; 3National Center for Biodefense and Infectious Diseases, George Mason University, Manassas, VA 20110, USA; fkashanc@gmu.edu; 4Department of Biochemistry and Molecular Biology, University of Miami Miller School of Medicine, Miami, FL 33136, USA; MToborek@med.miami.edu

**Keywords:** Zika virus, autophagy-defective mouse-model, Beclin1, microcephaly, inflammatory molecules, growth factor, congenital syndrome

## Abstract

Here, we used a mouse model with defective autophagy to further decipher the role of Beclin1 in the infection and disease of Zika virus (ZIKV)-R103451. Hemizygous (*Becn1*^+/−^) and wild-type (*Becn1*^+/+^) pregnant mice were transiently immunocompromised using the anti-interferon alpha/beta receptor subunit 1 monoclonal antibody MAR1-5A3. Despite a low mortality rate among the infected dams, 25% of *Becn1*^+/−^ offspring were smaller in size and had smaller, underdeveloped brains. This phenotype became apparent after 2-to 3-weeks post-birth. Furthermore, the smaller-sized pups showed a decrease in the mRNA expression levels of insulin-like growth factor (IGF)-1 and the expression levels of several microcephaly associated genes, when compared to their typical-sized siblings. Neuronal loss was also noticeable in brain tissues that were removed postmortem. Further analysis with murine mixed glia, derived from ZIKV-infected *Becn1*^+/−^ and *Becn1*^+/+^ pups, showed greater infectivity in glia derived from the *Becn1*^+/−^ genotype, along with a significant increase in pro-inflammatory molecules. In the present study, we identified a link by which defective autophagy is causally related to increased inflammatory molecules, reduced growth factor, decreased expression of microcephaly-associated genes, and increased neuronal loss. Specifically, we showed that a reduced expression of Beclin1 aggravated the consequences of ZIKV infection on brain development and qualifies *Becn1* as a susceptibility gene of ZIKV congenital syndrome.

## 1. Introduction

Zika virus (ZIKV) is a neurotropic flavivirus primarily transmitted by the Aedes mosquito [1,2,3]. Individuals infected with the virus typically develop mild symptoms. However, in utero, ZIKV exposure can cause congenital malformations, including microcephaly [4,5], and/or other overt congenital abnormalities, including fetal death, placental insufficiency, fetal growth restriction, and central nervous system injury [6]. Microcephaly is a neurodevelopmental disorder, characterized by a reduced head size when compared to babies of the same sex and age. The significant reduction in brain size, accompanied by intellectual disability, is believed to be caused by impaired cell proliferation and the death of both cortical progenitor cells and their neuronal progeny [7]. Significant downregulations of microcephaly-associated genes were detected in ZIKV-related studies [8,9,10], suggesting a direct mechanistic link of ZIKV infection to microcephaly. Twelve microcephaly genes, microcephalin-1 (MCPH1)-MCPH12, have been mapped [11]; many of which encode for proteins localized at the centrosome or proteins associated with centrosomal-related activities. These proteins play an important role in cell cycle progression, cell division, and formation of the mitotic spindle [12]. Furthermore, polyfunctional immune activation associated with an increased expression of cytokines, chemokines and growth factors has also been associated with neuronal damage, differentiation and proliferation of neural progenitor cells and fetal development [13].

Since the outbreak of ZIKV in 2015–2016, most reported animal studies are based on viral strains that were recovered from French Polynesia, Brazil, and other parts of South America; with only few reported cases on ZIKV strains recovered from Central America. Recently, we showed that ZIKV from both African and Asian lineages can modulate the autophagy pathway in glia (astrocytes and microglia) by silencing the autophagy gene, *Becn1*, which ultimately leads to increased inflammation in ZIKV-infected glia [14]. Beclin1 is a component of the phosphatidylinositol 3-kinase nucleation complex which regulates the initiation stages of the autophagy pathway [15]. Here, we used a hemizygous (*Becn1*^+/−^) and wild-type (*Becn1*^+/+^) mouse model to further decipher the role of autophagy in the infection and pathology of the epidemic human ZIKV strains, 2015/Honduras (R103451), during pregnancy and birth. Irrespective of murine genotype, infected dams showed high survival rates, along with high viral titers in various organs and in the placenta. On the other hand, pups delivered from ZIKV-infected *Becn1*^+/−^ dams showed significant growth impairments in their body and brain. Furthermore, in both genotypes, the downregulation of insulin-like growth factor (IGF)-1 and several microcephaly genes, along with neuronal loss, were detected in postmortem brain recovered from pups exposed to ZIKV. However, the reduced expression of Beclin1 aggravated the consequences of ZIKV infection. Further analysis with murine mixed glia, derived from ZIKV-infected *Becn1*^+/−^ and *Becn1*^+/+^ pups, showed greater infectivity in glia derived from the *Becn1*^+/−^ genotype, along with a significant increase in pro-inflammatory molecules, further supporting an association between Beclin1 and ZIKV.

## 2. Materials and Methods

### 2.1. Viral Propagation

Vero cells (Catalog# CRL-1586), mosquito cell line C6/36 (Catalog# CRL-1660), and the Honduran-R103451 (Catalog# VR-1848) ZIKV were procured from American Type Culture Collection (ATCC, Manassas, VA, USA). Vero cells or C6/36 were infected at a multiplicity of infection (MOI) of 0.01 for virus propagation, as described previously [14]. 

### 2.2. Ethics Statement

Animal work was conducted in accordance with the guidelines of the National Institutes of Health Guide for the Care and Use of Laboratory Animals. Animal experiments and associated protocols were reviewed and approved by the Florida International University Institutional Animal Care and Use Committee (IACUC# 17-007; 03/15/2018).

### 2.3. Animal Model and Timed Pregnancy 

*Becn1*^+/−^ (stock# 018429) hemizygous mice and C57BL/6J (stock# 000664) wild-type mice were procured from the Jackson Laboratory (Bar Harbor, ME, USA) and used exclusively for breeding in the animal facility at Florida International University. The *Becn1*^+/−^ mice were used as they express about 60% less Beclin1 protein and serve as a valuable tool to analyze the function of Beclin1 [16]. For the timed pregnancy studies, male and female mice aged 12 weeks were kept in isolation for at least 24 h prior to mating. In one set of experiments, *Becn1*^+/+^ sire were crossed with *Becn1*^+/+^ dam and delivered pups that were 100% homozygous for the *Becn1* allele and have a black coat color. In another set of experiments, *Becn1*^+/−^ sire were crossed with *Becn1*^+/−^ dam, and, of the pups delivered, approximately 2/3 were heterozygous for the *Becn1* gene and have an agouti coat color, believed to be a result of the effect of the *Becn1* mutation on melanogenesis [17]. Approximately 1/3 of the pups delivered were homozygous (*Becn1*^+/+^) with a black coat color [16,17]. Males and females were placed together in the early evening and monitored periodically for up to 48 h until the detection of a vaginal plug. The day that a plug was observed was considered embryonic or gestational day zero (E0). Pregnant dams received the anti-interferon (IFN) alpha/beta receptor subunit 1 monoclonal antibody MAR1-5A3, Leinco Technologies, MO, USA) at 2 mg/animal via intraperitoneal (ip) route at gestational day 8, followed by subcutaneous (sc) infection with ZIKV at 10^3^ plaque-forming unit (PFU) in 50 µL of PBS or mock (PBS) injection at the gestational day of E9. A booster dose of MAR1-5A3 at 0.5 mg/animal dose was administered by ip at 2- and 4 days post-infection (dpi). At E13 (four days after ZIKV challenge), maternal blood was collected, and serum was prepared after coagulation and centrifugation. At E17, dams were sacrificed and organs, including the brain, liver, heart and spleen, the placenta and fetuses were recovered. Organs were weighed and homogenized using a bead-beater apparatus (MagNA Lyser, Roche, Indianapolis). In parallel studies, at E20–E21, pups delivered were monitored for growth and weight changes for up to 3 weeks of age. After three weeks, pups born to ZIKV-infected or mock-exposed dams were sacrificed and brains removed postmortem were snap-frozen in liquid nitrogen for further use. Half of the brain hemisphere was minced and used for reverse transcription polymerase chain reaction (RT-PCR) analysis and the other half was used for histology and immunofluorescence staining as described below and in the accompanied Appendix A.

### 2.4. Real Time PCR

Viral RNA from serum and cellular RNA from tissues collected at various time-points from both ZIKV-infected and uninfected animals were extracted using QIAamp Viral RNA mini kit and RNeasy mini kit, respectively (Qiagen, Valencia, CA, USA). The extracted RNA was amplified by iTaq universal SYBR Green one-step PCR kit (Bio-Rad, Hercules, CA, USA) and 10 μM primers (Sigma-Aldrich, MO, USA). Expression of microcephalin-1 (MCPH1), WD repeat containing protein 62 (WDR62), cancer susceptibility candidate 5 (CASC5), and the abnormal spindle like primary microcephaly (ASPM) were measured using 500 ng of RNA extracted from pup brains. A standard curve was prepared from a 10-fold dilution of previously quantified ZIKV stock solution with known titer and viral titer expressed as Viral RNA (using standard curve). RT² Profiler™ PCR Array Mouse Autophagy was purchased from Qiagen (Catalog# PAMM-084Z). RNA extracted from the brain of pups born to infected dams were analyzed for the expression of autophagy-related genes following the manufacturer’s instruction and as previously described [14].

### 2.5. Primer Sequences

ZIKV:

     Forward 5′-CCGCTGCCCAACACAAG-3′

     Reverse 5′-CCACTAACGTTCTTTTGCAGACAT-3′

*MCPH1*:

     Forward 5′-AAGAAGAAAAGCCAACGAGAACA-3′

     Reverse 5′-CTCGGGTGCGAATGAAAAGC-3′

*ASPM*:

     Forward 5′-CCGTACAGCTTGCTCCTTGT-3′

     Reverse 5′-GGCGTTGTCCAATATCTTTCCA-3′

*CASC5*:

     Forward 5′-TCGCTGAAGTGGAAACAGAAAC-3′

     Reverse 5′-TATCTGAGCAAGGGTCTCTGCG-3′

*WDR62*:

     Forward 5′-GCTGACAAATGGCAAGCTG-3′

     Reverse 5′-GATGGTCTTGAGGGGTTCCT-3′

### 2.6. Hematoxylin & Eosin (H&E)

After the age of three weeks, pups were sacrificed, and brain tissues were removed postmortem and subsequently embedded in optimal cutting temperature (OCT) compound. Cryostat sectioned slices of 10-micron thickness were stained with H&E as described previously [18]. Images were acquired using an inverted fluorescence microscope with a 560 Axiovision camera and 20× and 40× objectives (Zeiss, Germany).

### 2.7. Murine Mixed Glial Cell Culture

For primary murine glial culture, postnatal day 4–6 (P4–P6) *Becn1*^+/−^ and *Becn1*^+/+^ littermates were separated according to phenotypic coat color and sacrificed according to previously described IACUC guidelines [16,19]. Cells seeded in 6-well plates were infected with ZIKV at an MOI of 0.1 or exposed with culture media, ZIKV envelope (E) and the non-structural protein (NS)-1 proteins. Viral proteins were purchased from ImmunoDx, Woburn, MA, USA. The protein concentration used (50 nM) was based on a dose–response curve and concentrations reported in cerebral spinal fluid (CSF) of patients with flavivirus infection [20].

### 2.8. ELISA

Cell culture supernatants (pre-cleared by brief centrifugation) were used to measure the levels of interleukin (IL)-6, monocyte chemotactic protein-1 (MCP-1), regulated on activation, normal T cell expressed and secreted (RANTES), and tumor necrosis factor alpha (TNF-α) using ELISA (R&D Systems, Minneapolis, MN, USA) according to the manufacturer’s instructions. The optical density was read at A450 on a Synergy HTX plate reader (BioTek, Winooski, VT, USA).

### 2.9. Plaque Assay

Vero cells were infected with a 10-fold dilution of ZIKV stock or the supernatants from infected/treated cells. After 1 h, adsorption cells were washed with PBS. The cells were overlaid with culture media (EMEM supplemented with 2% FBS) containing an equal volume of 3.2% carboxymethylcellulose and incubated for 5 days at 37 °C. The cells were fixed and stained with 1% crystal violet solution prepared in 20% formaldehyde, 30% ethanol and 50% PBS for 1 h. Stained cells were washed with water to remove excess crystal violet, left to dry overnight, and lysis plaques were quantified by stereomicroscope (Zeiss). The viral titer was expressed as plaque-forming units (PFU) per ml of the stock [14].

### 2.10. Immunohistochemistry

ZIKV infectivity was measured by fluorescent immunolabeling as described by Ojha et al. [14]. Briefly, the cells were fixed in 4% paraformaldehyde, permeabilized with 0.1% Triton X-100, and blocked in 10% milk/0.1% goat serum. The cells were immunolabeled with the astrocytes marker, Glial Fibrillary Acidic Protein (GFAP) antibody (Catalog# Ab7260; Abcam, Cambridge, MA, USA) and the anti-ZIKV-NS1 antibody (Catalog# GTX133307; Genetex, CA, USA). Immunoreactivity was visualized with secondary antibodies from Molecular Probes (Carlsbad, CA, USA). 4′,6-diamidino-2-phenylindole (DAPI) was used to label cell nuclei. Images were analyzed using an inverted fluorescence microscope with a 560 Axiovision camera (Zeiss).

### 2.11. Western Blotting

Protein was extracted from postmortem brain tissues of both *Becn1^+/+^* and *Becn1^+/−^* animals using Radioimmunoprecipitation assay (RIPA) buffer (Thermo Scientific, Waltham, MA, USA) supplemented with a mixture of protease and phosphatase inhibitors followed by SDS-PAGE protein separation. Immunoblots were labeled with primary antibodies against Beclin1 (Catalog# NB500–249), ATG5 (Catalog# NB110-53818), LC3-B (Catalog# NB600–1384) and P62/SQSTM1 (Catalog# NBP1–48320) purchased from Novus Biologicals (Centennial, CO, USA). β-actin (Catalog# sc-47778; Santa Cruz Biotechnology, Santa Cruz, CA, USA) was used as internal control. Immunoblots were subsequently incubated with secondary antibodies conjugated to horseradish peroxidase (Millipore, Billerica, MA, USA), exposed to SuperSignal West Femto Substrate (Thermo Scientific) and visualized using a ChemiDoc imaging system (Bio-Rad, Hercules, CA, USA). Densitometric analysis was quantitatively measured using image J (NIH.gov).

### 2.12. Statistical Analysis

The results are reported as the mean ± SEM of 3–5 independent experiments. The data were analyzed using analysis of variance (ANOVA) followed by the post hoc test for multiple comparisons (GraphPad Software, Inc., La Jolla, CA, USA). An alpha level (*p*-value) of <0.05 was considered significant.

## 3. Results

### 3.1. Transiently Immunosuppressed Pregnant Becn1^+/+^ and Becn1^+/−^ Dams Are Susceptible to ZIKV Infection

We explored the role of Beclin1 in ZIKV infection and disease using timed pregnancy in reduced Beclin1-expressing (*Becn1*^+/−^) and wild-type (*Becn1*^+/+^) mice models. For the in vivo studies, a schematic illustration of the experimental design is shown (Figure 1A), and further described in the Materials and Methods. The genotype of each animal strain was confirmed by PCR [16], followed by the detection of protein expression levels by Western Blotting (Figure 1B,C). Representative immunoblots confirmed a decrease in Beclin1 and LC3-II expression levels and increased p62/SQSTM1 levels in tissues extracted from *Becn1*^+/−^ mice when compared to *Becn1*^+/+^ mice (Figure 1B,C). The weight of each pregnant animal was measured before the detection of a vaginal plug and throughout the gestation period. Increases in body weight served as a measurable indicator of pregnancy among dams (Figure 1D). Beclin1 reduced (*Becn1*^+/−^) animals showed less gain in body weight compared to wild-type (*Becn1*^+/+^) dams. This was likely because *Becn1*^+/−^ delivered fewer pups when compared to *Becn1*^+/+^ dams, since litter numbers delivered by *Becn1*^+/−^ dams (crossed with an *Becn1*^+/−^ sire) are controlled by their genetic background. Litter numbers ranged between five and seven pups for *Becn1*^+/−^ dams and between six and nine pups for *Becn1*^+/+^ dams. Linear regression models (based on weight change from day (0) demonstrated that maternal weight gain at day 11 was a significant predictor of litter size (Figure 1D). The survival rates in pregnant dams’ post-infection with ZIKV was also monitored for the duration of gestation and showed minimal differences between mock- (PBS) and ZIKV-infected *Becn1*^+/+^ dams when compared to similar treated *Becn1*^+/−^ dams (Figure 1E). To confirm infection, serum was removed at gestation day E13 (4 days post-infection) and viral RNA was measured by RT-PCR. Viral RNA levels in the range of Log10^3^ PFU/mL were detected in the serum of *Becn1*^+/+^ and *Becn1*^+/−^ dams infected with ZIKV (Figure 1F). At E17 (8 days post-infection), maternal placenta and other organs removed postmortem from *Becn1*^+/+^ and *Becn1*^+/−^ dams, showed a high level of viral RNA in the placenta, followed by the spleen, liver, heart and the lowest titer was detected in the brain, irrespective of mice strain. The low level of viral RNA detected in the brain is indicative that ZIKV can cross the blood–brain barrier (Figure 1G). Overall, the data shows significant infection throughout gestation in both *Becn1* reduced and *Becn1*^+/+^ pregnant dams infected with ZIKV.

### 3.2. Growth Impairment in Pups Exposed to ZIKV In Utero

Vertical transmission of ZIKV from the placenta to the fetus was evident in embryos harvested at E17 (Figure 2A). In parallel studies, at E20–E21, pups born to mock (PBS) and ZIKV-infected *Becn1*^+/+^ and *Becn1*^+/−^ dams were monitored for up to 21-days for mortality and for morphological abnormalities. When compared to mock-infected animals, a slight decrease in the survival rate was noted in pups born to ZIKV-infected *Becn1*^+/+^ and *Becn1*^+/−^ (Figure 2B). A representative image of a litter born to ZIKV-infected *Becn1*^+/-^ dam is illustrated in Figure 2C, that consisted of both *Becn1*^+/+^ (black) and *Becn1*^+/−^ (agouti) pups. Within a litter, the smaller pup is indicated by a circle at day 7 and with an arrow at day 10. Fourteen days post-birth, growth abnormalities became exceedingly visible by differences in body size and was detected in 1 of every 4 (25%) pups. Genotyping confirmed that the smaller sized pups were heterozygous for the *Becn1* gene (data not shown). The average body weight was approximately 6.93 gm (Figure 2D) and the average body length was around 5.38 cm (Figure 2E). After 21 days, both small and typical sized pups were sacrificed, and brains were removed for further analysis. Viral proteins, NS1 and E, were detected in the brains of the 3-week-old pups (Appendix A). Representative images of 3-week-old pups born from ZIKV-infected and mock infected dams are shown in Figure 2F,G, respectively. Respective skull and brain images are shown on the right-hand side. The weight (in milligrams) of each brain determined by a scale is represented in a bar graph (Figure 2H) and the brain weight of the two groups within the *Becn1*^+/−^ genotype is represented in a chart in Figure 2I. The weight of the smaller brains was less than the weight of the well-defined brains. Overall, the data shows a growth dysfunction in pups born to ZIKV, reflected by decreases in body weight, body length, and brain weight. The impairment in body growth, along with abnormal brain morphology, was higher among pups born to *Becn1*^+/−^ mice infected with ZIKV.

### 3.3. Potential Factors Mediating the Pathology in Pups Exposed to ZIKV In Utero

Potential causal factor(s) responsible for the growth impairment detected in pups born to ZIKV-R103451-infected *Becn1*^+/−^ dams was further explored. The IGF-1, a polypeptide hormone with critical roles in regulating brain plasticity mechanisms, was reduced by 8-fold in *Becn1*^+/−^ pups born to ZIKV-infected dams when compared to *Becn1*^+/−^ pups born to mock-infected dams. On the contrary, IGF-1 was reduced by 4-fold in *Becn1*^+/+^ pups born to ZIKV-infected dams when compared to *Becn1*^+/+^ pups born to mock-infected dams (Figure 3A), suggesting a potential link between IGF-1 and ZIKV-associated growth impairments. The transmembrane protein 74 (TMEM74), a novel autophagy-related protein [21], was upregulated by approximately 6-fold in *Becn1*^+/−^ pups born to ZIKV-infected dams. Additional genes involved in the autophagy machinery are also included in the graph, although no significant differences were detected between *Becn1*^+/+^ and *Becn1*^+/−^ pups (Figure 3A). The expression levels of several microcephaly-associated genes, whose decrease in expression has been previously linked to stillbirth, brain development, and microcephaly in fetuses [8,22,23,24], were also measured by RT-PCR. The gene expression levels of MCPH1 and ASPM in brain tissues of *Becn1*^+/−^ pups born to mock (PBS)-exposed dams were significantly reduced when compared to *Becn1*^+/+^ pups born to mock-exposed dams (Figure 3B). Likewise, the expression levels of MCPH1, ASPM, CASC5 and WDR62 in brain tissues of *Becn1*^+/−^ pups born to ZIKV-infected dams were significantly lower (approximately 2.5–3-fold) when compared to *Becn1*^+/+^ pups born to ZIKV -infected dams (Figure 3B). The H&E staining of neurons showed no difference between mock-infected brains (Figure 3C; top and bottom panels), while ZIKV induced moderate neuronal loss in *Becn1*^+/+^ (Figure 3D; top panel), and more severe loss in brain recovered from *Becn1*^+/−^ mice (Figure 3D; bottom panel). Cell counting showed a significantly reduced number of H&E-stained neurons in *Becn1*^+/−^ mice exposed with ZIKV, when compared to similar exposed neurons in *Becn1*^+/+^ mice (Figure 3E). Overall, the data show a decrease in the expression of growth factors, microcephaly-associated genes, and visible signs of reduced neurons in the brains recovered from pups born to ZIKV-infected dams.

### 3.4. Reduced Beclin1 Exacerbates Secretion of Inflammatory Molecules in ZIKV-Infected Glia In Vitro

Since glial cells are the most abundant cell types in the brain and the principal cell types involved in the release of neuroinflammatory molecules; they are frequently considered the culprit in many viral pathologies [25,26]. To assess the role of glial cells in ZIKV-induced brain damage, mixed glia (astrocytes and microglia) were isolated from whole brain of either *Becn1*^+/+^ or *Becn1*^+/−^ pups, as described previously [16], and were infected with and without ZIKV at an MOI of 0.1. Mixed glial cultures were permissive to infection, albeit higher number of plaques were detected in *Becn1*^+/−^ glia infected with ZIKV. Figure 4A shows a representative image of glia derived from *Becn1*^+/+^ and *Becn1*^+/−^ pups infected with and without ZIKV after 24-h, followed by immunofluorescent labeling with the antibody against GFAP (red), ZIKV NS1 (green) and DAPI nucleus (blue). Viral infection and PFU were analyzed by plaque assays, using supernatants collected at various time-points post-infection (Figure 4B). Next, the secretion of inflammatory molecules was measured by ELISA using supernatant from non-infected (media) and ZIKV-infected glia. Infection with ZIKV caused a significant increase in RANTES, MCP-1 and IL-6 at 24-h that was still detected after 48-and 72-h post-infection (Figure 4C). Twenty-four-hours post-infection with ZIKV, secretion of RANTES was increased by 2.5-fold, MCP-1 was increased by 1.4-fold, and IL-6 was increased by 1.6-fold in supernatant derived from *Becn1*^+/-^ infected glial cells when compared to glia derived from *Becn1*^+/+^ pups (Figure 4C). Overall, the data show the infection of ZIKV in murine-derived glia along with the secretion of inflammatory molecules. Furthermore, the high levels of viral RNA correlate with the increased secretion of inflammatory cytokines in the ZIKV-infected *Becn1*^+/−^ glia after 24 and 48-h post-infection.

### 3.5. ZIKV- Proteins NS1 and E Causes Increase in the Secretion of Inflammatory Molecules

To further decipher the viral proteins involved in the upsurge of inflammatory molecules potentially contributing toward neuroinflammation, cultured glia cells were incubated with 50 nM of recombinantly expressed NS1 and E proteins. This concentration was based on a dose–response curve (Appendix A) and the concentration of proteins reported in the cerebral spinal fluid of patients with flavivirus infection [20] and in the sera of dengue virus (DENV)-infected patients [27,28]. Direct exposure of murine glia to NS1 or the E protein caused secretion in inflammatory molecules, with the most pronounced effects observed with TNF-α expression in supernatant from *Becn1*^+/−^ glia (Figure 5A). The secretion of IL-6 (B), RANTES (C), and MCP-1 (D) were also detected; however, the differences were minimal and not significant (Figure 5B–D).

Taken together, our data point to a potential link between Beclin1 and the regulation of TNF-α, although more in-depth studies are needed to further confirm potential protein-protein interactions.

## 4. Discussion

In the present study, we reported, for the first time, that the epidemic human ZIKV strains, 2015/Honduras (R103451), infects transiently immunocompromised autophagy defective pregnant dams with reduced Beclin1 expression levels (Figure 1). The reduced expression of Beclin1 aggravated the consequences of ZIKV infection on brain development and qualifies *Becn1* as a susceptibility gene of ZIKV congenital syndrome. The impact of ZIKV infection on dams were detected at E13 in serum, at E17 in placenta, and in other organs removed postmortem. There was limited viral RNA detected in the brain, despite the use of an anti-interferon (IFN) alpha/beta receptor subunit 1 (IFNAR1) monoclonal antibody (Figure 1). Low viral RNA detection in the brain is not unusual, since a report by Cao et al., 2017, also reported low levels of viral titers (in the range of 10–100 FFU equivalent/g) in fetal brain infected with the Brazilian strain of ZIKV [29], while others have shown high lethality with the African strain, MR766 [30]. The mechanism by which ZIKV replicates and causes congenital neurological complications, is not well understood [31]. According to a recent review [32], there are over 50 amino acid differences between the African and Asian ZIKV strains located in the NS1 (R67S; position 863), NS2B (S41T; position 1417), and NS5 (M60V; position 2634) proteins [31,32]. Differences in amino acid, together with the number of glycosylation sites in viral proteins [33], could present putative mechanisms for the differences in infectivity and pathogenicity observed between the viral strains. In our study, placenta recovered from postmortem dams infected with the Honduran strain of ZIKV showed high viral RNA levels (Figure 1) and about 25% of the heterozygous *Becn1*^+/−^ pups showed growth impairment (Figure 2). Surprisingly, growth impairment was evident in the 3-week-old pups, despite a lack viral RNA, while, viral proteins, NS1 and E, were detected in the brains of the 3-week-old pups (Appendix A), which is not uncommon. As shown by others, in a panel of patient sera infected with DENV, the NS1 protein was detected even in the absence of viral RNA or in the presence of immunoglobulin M antibodies. NS1 circulation levels varied among individuals during the course of the disease, ranging from several ng/mL to several ug/mL [28]. In other viruses, the presence of viral protein in the absence of viral RNA was reported in serum recovered from HIV-positive subjects treated with antiretroviral drugs, implying that viral RNA can be suppressed below detection level, while maintaining detectable protein expression in leaky reservoirs [34].

ZIKV mediated the reduction in the expression of microcephaly genes that are directly involved in neuronal cell division and proliferation may also contribute to the impairment in brain development [24,35]. Others have shown that mutations in the human *WDR62* resulted in microcephaly and a wide spectrum of cortical abnormalities [36,37,38], while a loss in the WDR62 protein function in mice causes mitotic delay, the death of neuron progenitor cells, reduced brain size and dwarfism [38]. *CASC5* was shown to be involved in cell cycle and kinetochore formation during metaphase with mutation in this gene was also implicated in causing microcephaly [39]. Using mouse models of *Mcph1* mutations, it was shown that microcephaly can develop due to the premature differentiation of neurons [40]. Furthermore, gliosis and neuronal damage were previously associated with ZIKV-infected microcephaly brain [41]. In the present study, a decrease in the expression of microcephaly genes was also detected in brains of *Becn1*^+/+^ pups born to ZIKV-R103451-infected dams, which could explain the observed the loss in neurons in postmortem brains recovered from pups born to infected dams (Figure 3), while attenuated Beclin1 expression further exacerbated the pathology (Figure 3E). Beclin1 and the ultraviolet irradiation resistance-associated gene (UVRAG) are involved in both autophagy and centrosome stability and linked to ZIKV-mediated microcephaly [42,43] While the recently identified MCPH18, a phosphatidylinositol 3-phosphate-binding protein, functions as a scaffold protein for the autophagic removal of aggregated protein, suggesting a potential link of autophagy in the development of primary microcephaly [44]. Autophagy can be involved in regulating the replication of ZIKV in cells of the central nervous system [14,45,46,47,48]. In related studies published by others, an autophagy-deficient animal model lacking the Atg16L gene showed restricted ZIKV infection in placenta, with reduced ZIKV-mediated placental damage and reduced adverse fetal outcomes [29]. The reduction in ATG16L1 expression levels in pregnant dams or placental trophoblastic cells showed limited ZIKV burden, which contradicts our current studies, as we did not detect significant differences in viral RNA of ZIKV infected autophagy-reduced dams when compared to ZIKV-infected autophagy-competent dams (Figure 1). Although speculative, the discrepancy between ATG16L1 and Beclin1 knock down (used in our studies) may relate to the differential role of the specific protein in the autophagy pathway and how specific steps in autophagy influence the life cycle and pathology of ZIKV. It is also important to reiterate that the hemizygous dams still express partially functional Beclin1 protein. In a related study by Peng et al., they used ZIKV-infected human umbilical vein endothelial cells and showed that both the pharmacological inhibition of the autophagy pathway and genetic inhibition of the *Becn1* gene, significantly reduced viral production [49]. While they used an in vitro cell culture system, which may not necessarily translate with what is seen in vivo, it is clear that autophagy has the potential to modulate ZIKV replication; a critical role that could be dependent on the tissue tropism and disease [50].

As for our findings, further studies, including gene silencing and the protein overexpression of the microcephaly genes are needed to better understand and decipher the mechanism involved in the growth impairment detected in our animal model. Alternatively, the low expression of IGF-1 detected in the postmortem brains of pups heterozygous for the *Becn1* gene born to ZIKV-infected dams may have triggered neuronal loss and subsequently downregulated the microcephaly genes. The IGF system plays a central role in hormonal growth regulation and is responsible for normal fetal and postnatal growth. For more than 30 years, IGF has been available as a replacement therapy in growth hormone-deficient patients and for the stimulation of growth in patients with short stature of various causes [51]. In a case study, a disruption of the IGF system in patient was associated with microcephaly, growth retardation, and intellectual disability [52]. Using a mice model with IGF-1 gene knockout, animals were presented with microcephaly and demyelination in the whole brain [53], whereas the overexpression of IGF-1 was shown to cause macrocephaly [53]. The concentrations of IGF-1 in the cerebral spinal fluid have been correlated with brain growth in autistic children [54], while low values of IGF-1 have been reported in a number of serious neurologic diseases of children [55]. In our study, levels of IGF-1 were significantly reduced in *Becn1*^+/-^ brain recovered in pups at 3 weeks of age (Figure 3A); this may be another underlying factor associated with the phenotype detected in our in vivo infectious model, while autophagy is required for proper functionality [56]. On the contrary, the expression of the TMEM74 was detected in brain tissue recovered from pups at 3 weeks of age (Figure 3A). TMEM74-related autophagy is independent of Beclin1/PI3KC3 complex, which may explain the reason that this gene was more expressed in animals lacking the *Becn1* gene and also, in the context of our animal model, may not be linked to ZIKV exposure [21]. Overall, the data show a decrease in the expression of growth factors, microcephaly genes and with visible signs of reduced neurons in brain recovered from pups born to ZIKV-infected dams; these factors may or may not be associated with the observed morphological abnormalities.

Recently, we showed the penetration of ZIKV across the brain parenchyma early after infection with concurrent alterations of tight junction protein expression and the disruption of the blood–brain barrier permeability [57]. The presence of viral proteins in the central nervous system can cause neuroinflammation, glial dysfunction, excitotoxicity, and neuronal death [45,58]. Glia have been found to play key roles in neuroinflammation, and although this is a normal and necessary process, emerging evidence in animal models suggests that sustained inflammatory responses by glia can contribute to disease progression [58] and possibly considered as a general underlying factor associated with the phenotype detected in our in vivo infectious model. Further analysis with murine mixed glia derived from *Becn1*^+/−^ and *Becn1*^+/+^ pups infected with ZIKV and exposed to viral proteins showed a significant increase in viral replication, which correlated with an increase in viral-induced cytokine and chemokine (Figure 4), thus supporting a link between Beclin1 and ZIKV infection.

Taken together, our results show growth impairment in body along with the abnormal brain morphology among pups born to *Becn1*^+/−^ mice infected with ZIKV, suggesting a potential function of Beclin1 in growth development. Our results also suggest that inflammatory molecules, growth factor and the autophagy machinery affect important aspects in the brain that could be associated with the neuronal loss and the growth impairment detected in ZIKV pathology. It is therefore of great significance to further investigate the role of autophagy in viral infection to address key issues regarding the mechanisms and treatment of ZIKV with the goal of identifying potential therapeutic interventions.

## Figures and Tables

**Figure 1 viruses-12-00608-f001:**
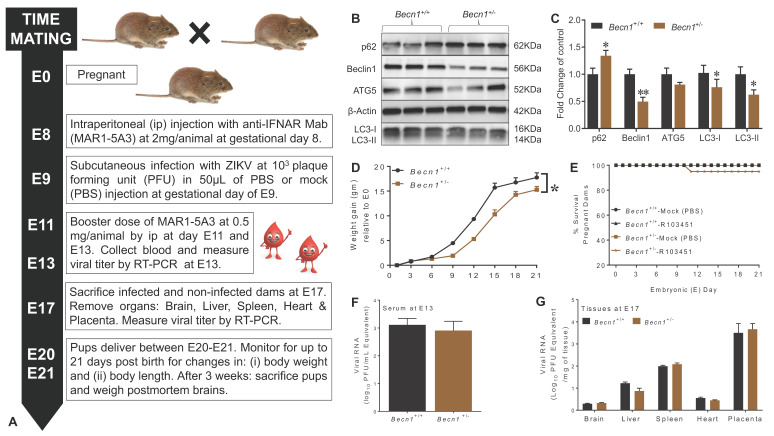
ZIKV infection in *Becn1*^+/+^ and *Becn1*^+/−^ pregnant dams. (**A**) Schematic diagram illustrating Zika virus (ZIKV)-infection in timed-pregnant dams. Prior to viral infection, pregnant dams received the antibody MAR1-5A3 at 2 mg/animal via intraperitoneal (ip) route at gestational day 8 followed by subcutaneous (sc) infection with ZIKV at 10^3^ plaque-forming unit (PFU) in 50 µL of PBS or mock (PBS) injection at gestational day E9. (**B**) Representative Western Blots probed with antibodies against several autophagy proteins, and B-actin was used as an internal control. Adult *Becn1*^+/+^ and *Becn1*^+/−^ brains were removed postmortem and minced according to the Materials and Methods. (**C**) Densitometric analysis using image J indicate the levels of p62, Beclin1, ATG5, LC3-I and LC3-II in brains of adult *Becn1*^+/+^ (black bar) and *Becn1*^+/−^ (brown bar) mice. The error bars show mean ± SEM for N = 3 animals per treatment. The data were analyzed using GraphPad Prism and two-way analysis of variance (ANOVA) followed by Tukey’s test. * *p* < 0.05 and ** *p* < 0.01 vs. *Becn1*^+/+^. (**D**) Weight gain, expressed in grams, was measured using an analytical balance at gestation day 0 and throughout gestation period, at 3-day intervals. (**E**) Percent survival rate in pregnant dams infected with ZIKV or mock (PBS) was calculated by dividing the total number of live animals by the number of live + dead animals X 100. (**F**) Viral RNA detected in serum collected from ZIKV-infected dams on E13. (**G**) Viral RNA detected in organs removed postmortem from ZIKV-infected dams on E17. (**D**–**G**) Error bars show mean ± SEM for *N* = 5–8 animals per treatment. The data were analyzed using GraphPad Prism and two-way ANOVA followed by Tukey’s test. * *p* < 0.05 vs. *Becn1*^+/+^. (**F**,**G**) Viral RNA equivalent is expressed on a log10 scale after comparison with a standard curve produced using serial 10-fold dilutions of ZIKV RNA from known quantities of infectious virus.

**Figure 2 viruses-12-00608-f002:**
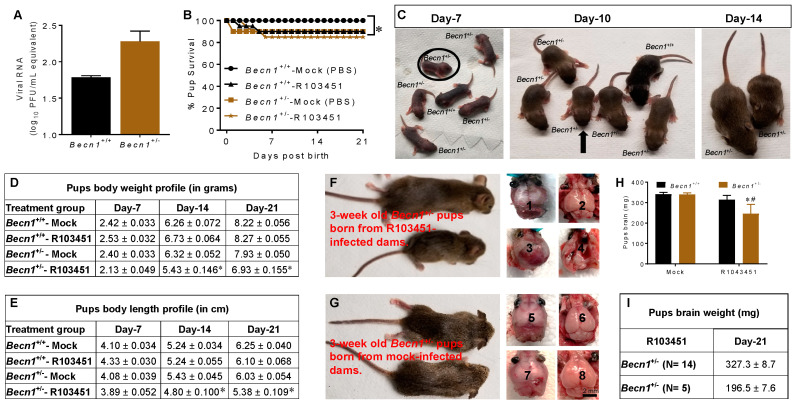
Growth impairment in pups born to ZIKV-R103451-infected dams. (**A**) Viral RNA was measured in postmortem fetuses collected on E17 from *Becn1*^+/+^ and *Becn1*^+/−^ dams infected with ZIKV. Viral RNA is expressed on a log10 scale after comparison with a standard curve produced using serial 10-fold dilutions of ZIKV RNA from known quantities of infectious virus. (**B**) Percent survival rate in pups born to ZIKV-infected dams was calculated using the total number animals subtracted by the dead pups X 100. (**C**) Representative image of a litter born to ZIKV- *Becn1*^+/−^ dam containing both *Becn1*^+/+^ and *Becn1*^+/−^ pups. The smaller pup is shown in a circle at day 7, with an arrow at day 10, and, at day 14, the smaller sized pup becomes more noticeable when compared to the regular sized sibling. (**D**) Body weight was measured using a balance and expressed in grams. (**E**) Body length was measured using a caliper and expressed in centimeter. (**F**,**G**) Representative images of 3-week-old pups born from ZIKV-infected (top) and mock (bottom) infected dams. Respective skull and brain images are shown on the right-hand side. Brain recovered from the small pups born to ZIKV-infected dams are labeled 3 and 4, while brain from the typical sized pups born to ZIKV-infected dams are labeled 1 and 2. Brain recovered from the typical sized pups born to mock-infected dams are labeled 5-8. Scale bar = 2 mm. (**H**) The brains’ weight in milligrams for *Becn1*^+/+^ and *Becn1*^+/−^ pups are represented in bar graph. (**I**) The brains’ weight in milligrams for the two groups within the *Becn1*^+/−^ pups are represented in a chart. (**A**,**B**,**D**,**E**,**H**) The error bars show the mean ± SEM of three experiments with *N* = 21 for *Becn1*^+/+^ and *N* = 19 for *Becn1*^+/−^ pups. The data were analyzed using two-way ANOVA followed by Tukey’s test for multiple comparison. * *p* < 0.05 vs. mock-infected *Becn1*^+/+^ strain, whereas, # *p* < 0.05 vs. mock infected *Becn1*^+/−^ strain.

**Figure 3 viruses-12-00608-f003:**
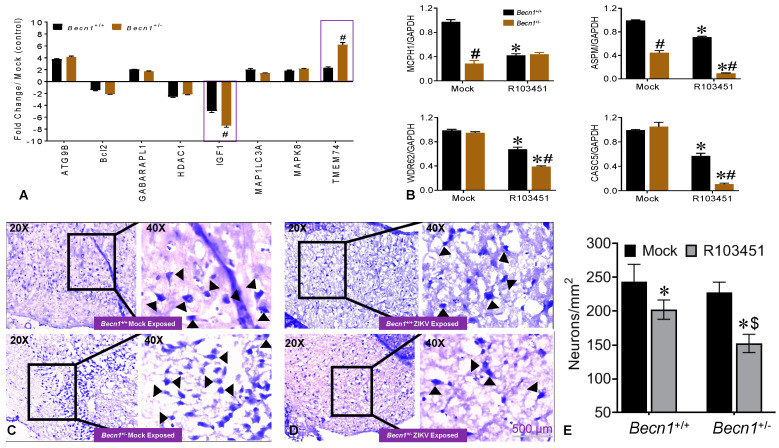
Reduced expression of growth factor, microcephaly-associated genes and neuronal loss in brain of pups exposed in-utero to ZIKV. (**A**) Half of the brain tissues were minced and used to measure autophagy related genes and growth factors in *Becn1*^+/+^ (black bar) and *Becn1*^+/−^ (brown bar) pups exposed to ZIKV in utero, by RT-PCR. The results are expressed as fold change from respective (mock) control (increase or decrease). (**B**) RNA expression of the microcephaly genes, microcephalin-1 (MCPH1), abnormal spindle like primary microcephaly (ASPM), WD repeat containing protein 62 (WDR62), and cancer susceptibility candidate 5 (CASC5) were measured by RT-PCR in brain of *Becn1*^+/+^ (black bar) and *Becn1*^+/−^ (brown bar) pups exposed to mock and ZIKV in utero. Expression levels are relative to *Becn1*^+/+^ pups born from wild-type mice and normalized to Glyceraldehyde 3-phosphate dehydrogenase (GAPDH). (**A**,**B**) The error bars show the mean ± SEM of three experiments with *N* = 21 for *Becn1*^+/+^ and *N* = 19 for *Becn1*^+/−^ pups. The data were analyzed by two-way ANOVA followed by Tukey’s multiple comparison test. * *p* < 0.05 vs. respective mock infected strain, ^#^
*p* < 0.05 vs. *Becn1*^+/+^. (**C**,**D**) Hematoxylin and Eosin (H&E) staining of three-week-old pup brains removed postmortem exposed to mock (left panel) and ZIKV (right panel) in utero. Images were acquired using an inverted fluorescence microscope with a 560 Axiovision camera using 20× and 40× magnification (Zeiss). Neurons are indicated by black arrows; scale bar = 500 µm. (**E**) Neurons were counted manually for mock and ZIKV treatment and averaged based upon the brain areas captured (mm2). The error bars show the mean ± SEM of three experiments with *N* = 21 for *Becn1*^+/+^ and *N* = 19 for *Becn1*^+/−^ pups. The data were analyzed by two-way ANOVA followed by Tukey’s multiple comparison test. * *p* < 0.05 vs. respective mock infected strain, ^#^
*p* < 0.05 vs. *Becn1*^+/+^.

**Figure 4 viruses-12-00608-f004:**
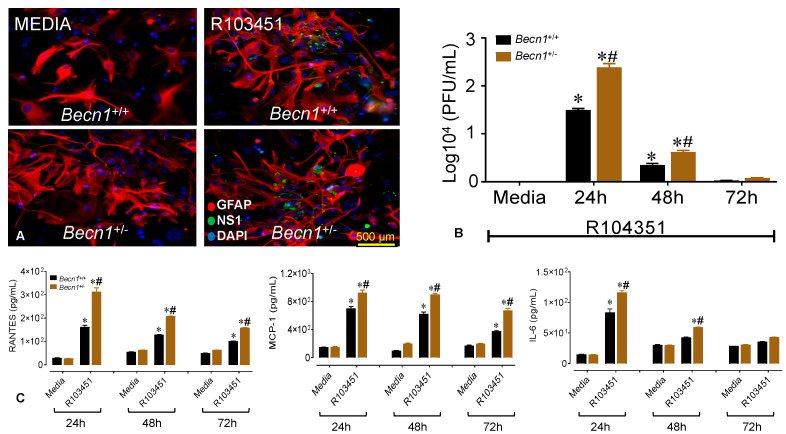
ZIKV infects mixed mouse glia and induces inflammatory molecules. (**A**) Representative immunofluorescent images of mouse mixed glia derived from *Becn1*^+/+^ (top) and *Becn1*^+/−^ (bottom) pups exposed to media (left) and infected with ZIKV (right). Cells were labeled with the antibody against Glial Fibrillary Acidic Protein (GFAP, red), the antibody against ZIKV NS1 (green), and 4′,6-diamidino-2-phenylindole (DAPI) was used to label nuclear DNA (blue). The images were acquired using an inverted fluorescence microscope with a 560 Axiovision camera and a 40× magnification (Zeiss). Scale bar = 500 µm. (**B**) Viral infection and PFU were analyzed by plaque assays at a dilution of 10^4^. The graph shows *Becn1*^+/+^ glia in black bar and *Becn1*^+/−^ glia in brown bar. (**C**) The secretions of regulated on activation, normal T cell expressed and secreted (RANTES), monocyte chemotactic protein-1 (MCP-1) and interleukin (IL)-6 were detected in glial supernatants infected with ZIKV at 24, 48- and 72-h post-infection by ELISA. The error bars show the mean ± SEM of 3 independent experiments. The data were analyzed by two-way ANOVA followed by Tukey’s multiple comparison test. * *p* < 0.05 vs. respective media control, ^#^
*p* < 0.05 vs. *Becn1*^+/+^.

**Figure 5 viruses-12-00608-f005:**
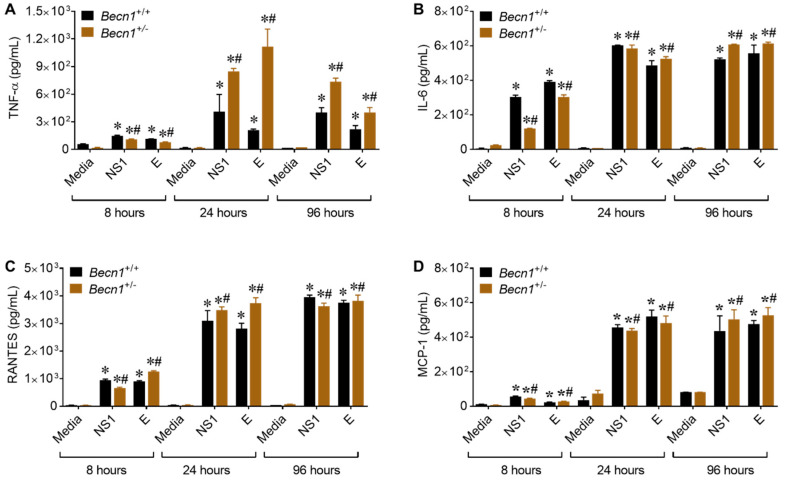
Potential link between ZIKV proteins and Beclin1 protein. (**A**–**D**) The secretions of pro-inflammatory molecules were detected in glial supernatants exposed to 50 nM of viral proteins after 8, 24- and 96-h by ELISA. *Becn1*^+/+^ glia (black bar) and *Becn1*^+/−^ glia (brown bar). The data were analyzed by two-way ANOVA followed by Tukey’s multiple comparison test. * *p* < 0.05 vs. respective media control, ^#^
*p* < 0.05 vs. *Becn1*^+/+^.

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
