# Peer review of "Reduced-Beclin1-Expressing Mice Infected with Zika-R103451 and Viral-Associated Pathology during Pregnancy"

_viruses, 2020, doi:10.3390/v12060608_

Round 1

Reviewer 1 Report

In this paper, Karuppan and colleagues investigate the consequences of reduced levels of Becn1 in female mice infected with Zika virus. They studied the impact of reduced autophagy on the progeny and measured brain size, the expression level of a few gene, viral replication in glia cells and the expression of a few inflammatory cytokines. They conclude that reduced autophagy is responsible for increased inflammatory molecules, reduced expression of "microcephalic genes" and neuronal loss. They state that autophagy could be a therapeutic target in ZIKV-induced growth abnormalities.

This paper presents a number of experiments which certainly deserve to be published. However, the manuscript must be revised for a number of reasons.

Major comments:

What is shown here is that reduced expression of Becn1 aggravates the consequences of ZIKV infection on brain development. It qualifies Becn1 as a susceptibility gene of ZIKV congenital syndrome. However, it does not show that ZIKV is acting through the autophagy machinery to induce microcephaly. Therefore, I do not share the final statement of the abstract on the potential therapeutic application of this finding. Likewise "reduced autophagy is related to ZIKV-induced pathology" (line 70) is not supported by the data.

The crosses are incompletely described, neither in the materials nor in the results sections. In such an experiment, it would be expected to cross +/- females with +/+ males to get both +/+ and +/- progeny which can be compared as littermates. However, when the authors report that +/- mothers gain less weight during pregnancy that +/+ females, they mention +/- x +/- crosses. Why? Did they also do crosses between +/+ females and +/- males? This is not mentioned. The genotype of the mother could influence the phenotype of the pups.

Move text to the right section: move experimental details from figure legends to materials section. Move interpretation of results to the discussion. Example: lines 292-297 belong to materials. Lines 303-306, 312 and 324-327 belong to discussion.

Apparently, microcephaly (threshold?) affected 30% of the +/- pups. Therefore, it would be more relevant to split the +/- pups in two groups according to their reduced body/brain weight, rather than showing raw means+/-sem which pool both groups.

The number of pups and litters analyzed is inconsistently reported. It must be indicated for each experiment and each panel.

The hypothesized interaction between Bcn1 and NS1 is not supported by the results. Figure 5B is unclear as currently shown.

Does the journal accept "data not shown " (4 instances)?

Becn1+/- mice are not "Beclin1-deficient" as declared in the title and in the text. Deficiency is defined by the complete absence, as in full knock-out strain. These mice have reduced activity (about 40%). This must be corrected.

Other comments:

"Viral RNA (PFU/mL)" is not correct. It should be "PFU/mL equivalent". Correct on all graphs.

The manuscript does not comply with the nomenclature rules. The official symbol for beclin 1 is Becn, not Atg6 (http://www.informatics.jax.org/mgihome/nomen/gene.shtml). Therefore, genotypes must be spelled Becn1+/+ (italicized) in every instance. Note that the protein must be written BECN1 (uppercase, not italicized).

"Microcephalic genes" (lines 29, 64, 325, 329, 447, 466, 469) is incorrect. These are genes which have been identified in human primary microcephaly (ref 11). "Microcephaly-associated genes" (line 44) is more appropriate. Microcephalin is of them (MCPH1). "Twelve of the microcephalin loci" (line 45) is incorrect.

Table 1: separate from Figure 1. Table 1 legend must not appear in Figure 1 legend. Show the genotypes of dam and sire. What days for blood collection and viral titer by RT-qPCR? Brain weight was not "monitored for up to 21 days".

Figure 1: Panel numbering does not match between Fig1 and its legend. C shows body weight up to 15 days in +/+ females and 16 days for +/-. Give N. Legend says every 3 days but graph is daily. To which comparison does * refer? E: give N. F: these are organ extracts. mL of what ? The detection threshold must be shown (brain values are shown to be under 0.3 PFU/mL!!). Several organs are certainly under this threshold.

Figure 2: A has no unit. B: any significant difference? E is useless (D is enough). H: instead of bars, show individual values to highlight the spread of values for the infected +/- pups.

Figure 3C: top-left panel is scrambled

Figure 4B: the insert showing plaques is useless. Very small error bars are surprising. Is each value a different biological replicate (different culture, different experiment)? B: express results directly in 10<6>.

Figure 5: what is really WB shown in B ? Where is the B-actin control ? Which Ab was used ? What does ZIKV mean ? Remove C which does not provide useful information and not discussed.

Line 17: what justify the term "translatable"?

Line 19: anti-IFNAR monoclonal AB. This Ab is MAR1-5A3 (not MAR-5A3).

Line 62-63: incomplete sentence

Line 66: "further reductions": unclear

Line 76: "for virus propagation"

Line 84: "and C57BL/6J (stock # 000664)"

Line 259: "was detected in 1 of every 4 (25-30%) pups". How was it detected? Were they weighted? 1 out of 4 is exactly 25%, not 25-30%. Give the precise numbers.

Line 260-261: give the age

Line 334: "by RT-PCR in the brain of"

Lines 364-365: move to discussion

Line 366: "high level of viral RNA measured". It was measured by plaque assay (line 357).

Line371: "infected with ZIKV"

Line 385: "As expected" is inappropriate. Same comment for "As a proof-of-concept". These are hypotheses under test.

Lines 409-411: this is wrong. These mice have been treated with MAR1-5A3 which renders them susceptible. It is certainly not the first time that B6 mice treated with this Ab are reported to be susceptible. There are several reports in the literature.

Line 417: reference missing.

Lines 424-429: ZIKV is rapidly eliminated during infection (undetectable after one week in adult mice). What is seen here are the consequences of an infection which occurred during development. This discussion is not relevant. Likewise, the discussion of lines 433-436 is not related to this study.

Lines 446-447: is there published evidence that a similar reduction in microcephaly-associated genes in a non-infectious context may result in neuronal loss?

Line 455: give a ZIKV-specific reference.

Line 465: gene silencing of which gene? Bcn1+/- mice are used here and Bcn1-/- are non viable.

Lines 469-481: Igf1 is obviously an important growth factor for brain development. This part of the discussion is not justified by the data shown.

Lines 490-491: "Although the in vitro data does not necessarily support the causal factors detected in the in vivo studies": this is a vague statement which weakens the outcome of this study.

Line 494: which experiment specifically addressed "neuron-glia communication"?

Author Response

We thank the reviewers for taking the time to read our manuscript and provide valuable comments and suggestions. The changes are marked in blue font below and tracked in the manuscript.

REVIEWER #1:

Q1:What is shown here is that reduced expression of Becn1 aggravates the consequences of ZIKV infection on brain development. It qualifies Becn1 as a susceptibility gene of ZIKV congenital syndrome. However, it does not show that ZIKV is acting through the autophagy machinery to induce microcephaly. Therefore, I do not share the final statement of the abstract on the potential therapeutic application of this finding. Likewise "reduced autophagy is related to ZIKV-induced pathology" (line 70) is not supported by the data.

Response: We thank the reviewer for his suggestion and have revised the final statement as follows, “In the present study, we identified a link by which defective autophagy is causally related to increased inflammatory molecules, reduced growth factor, decreased expression of microcephaly-associated genes, and increased neuronal loss. Specifically, we showed that reduced expression of Beclin1 aggravated the consequences of ZIKV infection on brain development and qualifies Beclin1 as a susceptibility gene of ZIKV congenital syndrome”.

Q2: Likewise "reduced autophagy is related to ZIKV-induced pathology" (line 70) is not supported by the data.

Response: The sentence on Line 70 has been removed.

Q3: The crosses are incompletely described, neither in the materials nor in the results sections. In such an experiment, it would be expected to cross +/- females with +/+ males to get both +/+ and +/- progeny which can be compared as littermates. However, when the authors report that +/- mothers gain less weight during pregnancy that +/+ females, they mention +/- x +/- crosses. Why? Did they also do crosses between +/+ females and +/- males? This is not mentioned. The genotype of the mother could influence the phenotype of the pups.

Response: We have added more information about the crosses in the materials and methods section. The paragraph reads as follows: “In one set of experiments, Becn1+/+ sire were crossed with Becn1+/+ dam and delivered pups that were 100% homozygous for the becn1 allele and have a black coat color. In another set of experiments, Becn1+/- sire were crossed with Becn1+/- dam and of the pups delivered were approximately 50% heterozygous for the becn1 gene and have an agouti coat color , believed to be a result of the effect of the Becn1 mutation on melanogenesis (17). About 25% of the pups delivered are homozygous (Becn1+/+) with a black coat color, while homozygous deletion of the targeted allele (Becn1-/-) resulted in embryonic lethality (16,17).   

Q4: Did they also do crosses between +/+ females and +/- males?

Response: We did not cross between +/+ females and +/- males. 

Q5: Move text to the right section: move experimental details from figure legends to materials section. Move interpretation of results to the discussion. Example: lines 292-297 belong to materials. Lines 303-306, 312 and 324-327 belong to discussion.

Response: We have moved lines 293-297 to materials and methods. Lines 303-306, 312 and 327 were moved to discussion.

Q6: Apparently, microcephaly (threshold?) affected 30% of the +/- pups. Therefore, it would be more relevant to split the +/- pups in two groups according to their reduced body/brain weight, rather than showing raw means+/-sem which pool both groups.

Response: We thank the reviewer for his/her suggestions and have replaced Fig. 2I with a chart that represents brain weight of the two groups within the +/- genotype.

Q7: The number of pups and litters analyzed is inconsistently reported. It must be indicated for each experiment and each panel.

Response: we have included the numbers analyzed for each experiment and each panel.

Q8: The hypothesized interaction between Bcn1 and NS1 is not supported by the results. Figure 5B is unclear as currently shown.

Response: we agree and have decided to remove Fig. 5B. 

Q9: Does the journal accept "data not shown " (4 instances)?

Response: Line 384 (data not shown) has been replaced with Supplemental Fig.1.

Line 389 (data not shown) has been replaced with Fig. 5B, C, D

Line 430 (data not shown) has been replaced with supplemental Fig. 2

Q10: Becn1+/- mice are not "Beclin1-deficient" as declared in the title and in the text. Deficiency is defined by the complete absence, as in full knock-out strain. These mice have reduced activity (about 40%). This must be corrected.

Response: we agree and have replaced the title as follows: REDUCED-BECLIN-1-EXPRESSING MICE INFECTED WITH ZIKA-R103451 AND VIRAL-ASSOCIATED PATHOLOGY DURING PREGNANCY.  In the text the  word “deficient” was replaced with “reduced”.

Other comments:

Q11: "Viral RNA (PFU/mL)" is not correct. It should be "PFU/mL equivalent". Correct on all graphs.

Response: PFU/mL has been corrected to PFU/mL equivalent

Q12: The manuscript does not comply with the nomenclature rules. The official symbol for beclin 1 is Becn, not Atg6 (http://www.informatics.jax.org/mgihome/nomen/gene.shtml). Therefore, genotypes must be spelled Becn1+/+ (italicized) in every instance. Note that the protein must be written BECN1 (uppercase, not italicized).

Response: we have replaced Atg6 with Becn1(italicized) in every instance

Q13: "Microcephalic genes" (lines 29, 64, 325, 329, 447, 466, 469) is incorrect. These are genes which have been identified in human primary microcephaly (ref 11). "Microcephaly-associated genes" (line 44) is more appropriate. Microcephalin is of them (MCPH1). "Twelve of the microcephalin loci" (line 45) is incorrect.

Response: The word microcephalic was replaced with microcephaly genes.  “Twelve of the microcephalin loci “ was replaced with:  “Twelve of the microcephaly genes, (MCPH) loci (MCPH1-MCPH12) have been mapped (11); many of which encodes for proteins localized at the centrosome or proteins associated with centrosomal-related activities …….”

Q14: Table 1: separate from Figure 1. Table 1 legend must not appear in Figure 1 legend. Show the genotypes of dam and sire. What days for blood collection and viral titer by RT-qPCR? Brain weight was not "monitored for up to 21 days".

Response: Table 1 was removed from Figure1 and now appears as a stand-alone Table.  Blood was collected at E5 (prior to infection) and at E13 (post-infection); as indicated by the blood drops. Brain weight was measured at day 21 after mice were sacrificed (the error was corrected in the Table). Genotype of the dams is reflected by the expression of Beclin1 and other autophagy related proteins in Fig. 1A and B.

Q15: Figure 1: Panel numbering does not match between Fig1 and its legend. C shows body weight up to 15 days in +/+ females and 16 days for +/-. Give N. Legend says every 3 days but graph is daily. To which comparison does * refer? E: give N. F: these are organ extracts. mL of what ? The detection threshold must be shown (brain values are shown to be under 0.3 PFU/mL!!). Several organs are certainly under this threshold.

Response: Fig. 1C was corrected. Error bars show mean ± SEM for N = 5 - 8 animals per treatment. The stats symbols refer to:  *p<0.05 and ** p<0.01 vs. Becn1+/+. Fig. 1F: Organ extracts are indicated as mg of tissue. We have corrected the unit for the y-axis.

Q16: Figure 2: A has no unit. B: any significant difference? E is useless (D is enough). H: instead of bars, show individual values to highlight the spread of values for the infected +/- pups.

Response: Unit has been added to Fig. 2A.  Symbol has been added to Fig.2B.  We appreciate the comment by the reviewer but prefer to include Fig. 2E.

Fig. 2H: We appreciate the suggestion, however, because of large number of pups, the values overlapped, making the figure less distinctive. We have replaced Fig. 2I with a chart that represents brain weight of the two groups within the Becn1+/- genotype.

Q17: Figure 3C: top-left panel is scrambled.

Response: The figure has been replaced

Q18: Figure 4B: the insert showing plaques is useless. Very small error bars are surprising. Is each value a different biological replicate (different culture, different experiment)? B: express results directly in 10<6>.

Response: We removed the plague assay.  The error bars are small because the values are expressed in logarithmic scale. Fig.4B is now expressed as Log10 (PFU/mL).

Q19: Figure 5: what is really WB shown in B ? Where is the B-actin control ? Which Ab was used ? What does ZIKV mean ? Remove C which does not provide useful information and not discussed.

Response: We have removed Fig. 5B and Fig. 5C.

Q20: Line 17: what justify the term "translatable"?.

Response: The term has been removed.

Q21: Line 19: anti-IFNAR monoclonal AB. This Ab is MAR1-5A3 (not MAR-5A3).

Response: We thank the reviewer for his/ her keen observation and Line 19 has been corrected to MAR1-5A3.  

Q22: Line 62-63: incomplete sentence

Response: Sentence was corrected, and now reads as follows” Irrespective of murine genotype, infected dams showed high survival rates, along with high viral titers in various organs and in the placenta. On the other hand, pups delivered from ZIKV-infected Becn1+/- dams showed significant growth impairments in their body and brain”.

Q23: Line 66: "further reductions": unclear

Response: The words were removed, and now reads as follows “Furthermore, in both genotypes, downregulation of insulin-like growth factor (IGF)-1 and several microcephaly genes, along with neuronal loss, were detected in postmortem brain recovered from pups exposed to ZIKV”.  

Q24: Line 76: "for virus propagation"

Response: Line 76 has been corrected

Q25: Line 84: "and C57BL/6J (stock # 000664)"

Response: Line 84 has been corrected

Q26: Line 259: "was detected in 1 of every 4 (25-30%) pups". How was it detected? Were they weighted? 1 out of 4 is exactly 25%, not 25-30%. Give the precise numbers.

Response: We have removed 30% and kept 25%. 1 in 4 pups had smaller body length and less body weight.

Q27: Line 260-261: give the age. 

Response: The pups were 3-weeks of age.  The sentence was revised as follows “ Genotyping confirmed that the smaller-sized pups were ……..

Q28: Line 334: "by RT-PCR in the brain of"

Response: Line 334 has been corrected

Q29: Lines 364-365: move to discussion

Response: The phrase” thus supporting a link between Beclin1 and ZIKV” has been moved to discussion

Q30: Line 366: "high level of viral RNA measured". It was measured by plaque assay (line 357).

Response: We agree with the reviewer and replaced the sentence with “higher number of plaques detected”

Q31: Line371: "infected with ZIKV"

Response: Line 371 has been corrected

Q32: Line 385: "As expected" is inappropriate. Same comment for "As a proof-of-concept". These are hypotheses under test.

Response: We have removed the words "As expected" and "As a proof-of-concept" from the manuscript.

Q33: Lines 409-411: this is wrong. These mice have been treated with MAR1-5A3 which renders them susceptible. It is certainly not the first time that B6 mice treated with this Ab are reported to be susceptible. There are several reports in the literature.

Response: We agree with the comment of the reviewer, what we meant to say was the following” In the present study, we reported for the first time that the epidemic human ZIKV strains, 2015/ Honduras (R103451), infects transiently immunocompromised autophagy defective pregnant dams with reduced Beclin1 expression levels. 

Q34: Line 417: reference missing.

Response: Reference has been added

Q35: Lines 424-429: ZIKV is rapidly eliminated during infection (undetectable after one week in adult mice). What is seen here are the consequences of an infection which occurred during development. This discussion is not relevant.

Response: Lines 424-429 has been removed

Q36: Likewise, the discussion of lines 433-436 is not related to this study.

Response: The discussion was in reference to other viruses (in addition to ZIKV) that can be detected by proteins without presence of RNA. 

Q37: Lines 446-447: is there published evidence that a similar reduction in microcephaly-associated genes in a non-infectious context may result in neuronal loss? 

Response: Others have examined the role several microcephaly genes, including ASPM, MCPH1 and CDK5RAP2 in the risk of Alzheimer disease (AD), i.e., PMCID: PMC3136560, PMID: 31788001.

Q38: Line 455: give a ZIKV-specific reference.

Response: References were added specific to ZIKV

Q39: Line 465: gene silencing of which gene? Bcn1+/- mice are used here and Bcn1-/- are non viable.

Response: We apologize for the confusion. Gene silencing was in references to the microcephaly genes.

Q40: Lines 469-481: Igf1 is obviously an important growth factor for brain development. This part of the discussion is not justified by the data shown.

Response: lines 469-481, was provided to highlight the importance of IGF in growth development and to provide scientific premise as shown by others that IGF is associated with microcephaly.  The information on lines 469-481 were provided to strengthen our findings that the reduced levels of IGF in Becn1+/- brain may be another underlying factor associated with the phenotype detected in our in vivo infectious model. 

Q41: Lines 490-491: "Although the in vitro data does not necessarily support the causal factors detected in the in vivo studies": this is a vague statement which weakens the outcome of this study.

Response: Statement was removed

Q42: Line 494: which experiment specifically addressed "neuron-glia communication"?

Response: We have replaced “neuron-glia communication” with the word “brain”; as this was the organ used in this study.

Reviewer 2 Report

In this work, Karuppan et al use BECN1-deficient mice to study Zika virus associated pathology. They confirm the smaller size as well as small and undeveloped brains in offspring of BECN1-deficient mice, suggesting an important role of autophagy during Zika virus induced microcephaly. The animal studies are supported by the analyses of relevant molecular markers; both are well presented in this manuscript. There are some suggestions, which would improve the quality of this work:

  • 1A. The presented WB (β-actin, lower panel) consists of 2 pieces, thus precluding reliable quantification and normalisation of the other genes.
  • 2F-G. The skull and brain images lack scale bar (ruler) to reliably compare the size of organs.
  • 3C-D. The H&E staining images should be complemented with Zika-specific antibody staining to confirm that the corresponding phenotypical changes correlate with the infection.
  • The experiment describing immunoprecipitation does not correspond to the provided image (Fig. 5B). As stated in Materials and Methods, immunoblots were probed with anti-Beclin1, anti-NS1, anti-E and anti-β-actin. Please provide all 4 images to support the conclusions.
  • The docking experiment is rather descriptive (Fig. 5C) and is not supported by any calculations (e.g. docking accuracy, predicted affinity and other scoring parameters).
  • While describing their findings, the authors ignore another similar work published earlier: Peng et al 2018 (PMID 29762492).

Author Response

We thank the reviewers for taking the time to read our manuscript and provide valuable comments and suggestions. The changes are marked in blue font below and tracked in the manuscript.

REVIEWER #2:

Q1: 1A. The presented WB (β-actin, lower panel) consists of 2 pieces, thus precluding reliable quantification and normalisation of the other genes.

Response: B-actin was used as internal control. The first three lanes represent adult brain recovered from becn1+/+ mice and lanes 4, 5 and 6  represent adult brain recovered from Becn1+/- mice. Protein from each lane was normalized to its respective B-actin.

Q2: 2F-G. The skull and brain images lack scale bar (ruler) to reliably compare the size of organs.

Response: A scale bar indicating 2 mm has been added to the images.

Q3: 3C-D. The H&E staining images should be complemented with Zika-specific antibody staining to confirm that the corresponding phenotypical changes correlate with the infection.

Response: Immunofluorescence images of Zika-specific antibody staining has been included in supplemental figure 1.

Q4: The experiment describing immunoprecipitation does not correspond to the provided image (Fig. 5B). As stated in Materials and Methods, immunoblots were probed with anti-Beclin1, anti-NS1, anti-E and anti-β-actin. Please provide all 4 images to support the conclusions.

Response: We have removed Fig. 5B.

Q5: The docking experiment is rather descriptive (Fig. 5C) and is not supported by any calculations (e.g. docking accuracy, predicted affinity and other scoring parameters).

Response: We agree with the reviewer and have removed Fig. 5C.

Q6: While describing their findings, the authors ignore another similar work published earlier: Peng et al 2018 (PMID 29762492).

Response: We apologize for this oversight and have cited the work by Peng et al., 2018 in the discussion part that reads as follows” In a related study by Peng et al., they used ZIKV-infected human umbilical vein endothelial cells (HUVEC) and showed that both pharmacological inhibition of the autophagy pathway and genetic inhibition of the BECN1 gene, significantly reduced viral production (49). While they used an in vitro cell culture system, which may not necessarily translate with what is seen in vivo, it is clear that autophagy has the potential to modulate ZIKV replication; a critical role that could be dependent on the tissue tropism and disease (50)”.

Round 2

Reviewer 1 Report

Most of the points I previously raised have been properly addressed.

There are however a few remaining corrections to make.

In the " 2.3. Animal model and timed pregnancy" section, the proportions from the +/- x +/- cross are wrong since Becn1-/- embryos die in utero. Therefore, out of the delivered pups, 2/3 are +/- and 1/3 are +/+.

Abstract : monoclonal antibody (not plural)

In 3.1 : "ranged between 5 to 7 pups for Becn1+/- FEMALES and between 6 to 9 pups for Atg6Becn1+/+ FEMALES.

The paragraph in 3.3 starting with "Using H&E staining" is confusingly written and must be rephrased more clearly. The main observations are that (1) there is no difference between mock-infected embryos; (2) ZIKV induced moderate neuronal loss in Becn1+/+ embryos, more severe in Becn1+/- embryos. Panels C and D in Figure 3 show only 2 of the 4 combinations, with same combination top and bottom pictures. Is it a labelling error? There MUST be 4 panels showing the 4 combinations (2 genotypes x mock or infected).

In 3.4, replace "To this end" with "To assess the role of glial cells in ZIKV-induced brain damage"

Figure 4B shows PFU/ml after a 10<4> dilution (see figure legend). Change Y-axis scale to show the actual values which will be much more meaningful. Amend figure legend accordingly.

Beginning of discussion : MAR1-5A3 is NOT an anti-interferon mAb. It is an anti-IFNAR mAb (directed against the receptor to type I IFNs).

Bottom of page 15 : Low VIRAL RNA detection

The sentence "According to a recent review..." does not refer to a specific reference.

"25% of the heterozygous Becn1+/- pups"

Please, check the correct spelling "Becn1" in italics with capital B at every instance when referring to the gene.

Author Response

We are extremely thankful to reviewer 1 for providing helpful comments and suggestions.  The changes are marked in blue font in the manuscript.

In the " 2.3. Animal model and timed pregnancy" section, the proportions from the +/- x +/- cross are wrong since Becn1-/- embryos die in utero. Therefore, out of the delivered pups, 2/3 are +/- and 1/3 are +/+.

Response: We have revised the statement and now reads: “In another set of experiments, Becn1+/- sire were crossed with Becn1+/- dam and of the pups delivered approximately 2/3 were heterozygous for the becn1 gene and have an agouti coat color, believed to be a result of the effect of the Becn1 mutation on melanogenesis (17). Approximately 1/3 of the pups delivered were homozygous (Becn1+/+) with a black coat color (16,17).

Abstract : monoclonal antibody (not plural)

Response: The word has been fixed.

In 3.1 : "ranged between 5 to 7 pups for Becn1+/- FEMALES and between 6 to 9 pups for Atg6Becn1+/+ FEMALES. 

Response: The sentence was corrected and reads as follows: “Litter numbers ranged between 5 to 7 pups for Becn1+/- dams and between 6 to 9 pups for Becn1+/+  dams”.

The paragraph in 3.3 starting with "Using H&E staining" is confusingly written and must be rephrased more clearly. The main observations are that (1) there is no difference between mock-infected embryos; (2) ZIKV induced moderate neuronal loss in Becn1+/+ embryos, more severe in Becn1+/- embryos. Panels C and D in Figure 3 show only 2 of the 4 combinations, with same combination top and bottom pictures. Is it a labelling error? There MUST be 4 panels showing the 4 combinations (2 genotypes x mock or infected).

Response: We thank the reviewer for this observation that was a labeling error on our part.  The mistake has been corrected.  The paragraph was rephrased as follows: “H&E staining of neurons showed no difference between mock-infected brains (Fig. 3C; top and bottom panels), while ZIKV induced moderate neuronal loss in Becn1+/+ (Fig. 3D; top panel), and more severe loss in brain recovered from Becn1+/- mice (Fig. 3D; bottom panel).  Cell counting showed a significantly reduced number of H&E-stained neurons in Becn1+/- mice exposed with ZIKV, when compared to similar exposed neurons in Becn1+/+ mice (Fig. 3E)”.

In 3.4, replace "To this end" with "To assess the role of glial cells in ZIKV-induced brain damage"

Response: The sentence was replaced as suggested by the reviewer.

Figure 4B shows PFU/ml after a 10<4> dilution (see figure legend). Change Y-axis scale to show the actual values which will be much more meaningful. Amend figure legend accordingly.

Response: The Figure and legend have been fixed accordingly.

Beginning of discussion : MAR1-5A3 is NOT an anti-interferon mAb. It is an anti-IFNAR mAb (directed against the receptor to type I IFNs).

Response: “anti-interferon mAb” has been replaced with “anti-IFNAR monoclonal antibody”

Bottom of page 15 : Low VIRAL RNA detection

Response: The word “viral” has been added.

The sentence "According to a recent review..." does not refer to a specific reference.

Response: Reference has been added.

"25% of the heterozygous Becn1+/- pups"

Response: Sentence has been corrected.

Please, check the correct spelling "Becn1" in italics with capital B at every instance when referring to the gene.

Response: Becn1 in italics has been added everywhere when referring to the gene.
